# An Introduction to Quantum Mechanics Through Neuroscience and CERN Data

**Héctor Reyes-Martín** [1],* and **María Arroyo-Hernández** [2]

1 Higher Polytechnic School, Universidad Francisco de Vitoria, 28223 Madrid, Spain
2 Experimental Science Faculty, Universidad Francisco de Vitoria, 28223 Madrid, Spain; m.arroyo.prof@ufv.es
* Correspondence: hector.reyes@ufv.es

**Abstract:** (1) Background: One of the greatest challenges students face when studying quantum mechanics is the lack of daily experience and intuition about its concepts. This article introduces a holistic activity designed to present some foundational ideas of quantum mechanics in a new pedagogical approach to enhance students' motivation. Using real open data from CERN, the activity connects classical concepts of dynamics and electromagnetism to their quantum counterparts, emphasizing both their similarities and differences. Teaching physics must consider the way the brain learns. That is why the activity is based on observed neuroscientific principles of physics learning. The approach maintains the rigor and precision required for these abstract concepts. (2) Methods: To evaluate the activity's impact by gender, intrinsic motivation was assessed using a Likert-type scale with 81 undergraduate students from fields including artificial intelligence systems engineering, computer engineering, mathematical engineering, and architecture. (3) Results: a Mann–Whitney U test analysis indicates the activity significantly enhances students' intrinsic motivation to study quantum mechanics, with improvements observed in both male and female students. (4) Conclusions: This result highlights the potential of the activity to promote greater interest in physics, both in men and women, since no significant differences have been observed between both samples.

**Keywords:** quantum physics learning; motivation; physical particles; women in physics; CERN; CMS; neuroscience

## 1. Introduction

Given the importance of quantum physics and its applications, its study is being introduced into high school curricula and introductory university courses in various countries [1]. However, quantum physics presents unique challenges that necessitate the development of new learning strategies.

Quantum physics has earned a reputation for being particularly difficult for students of physics, and there are valid reasons for this. The most significant challenges include the mathematical formalism, interpreting non-intuitive phenomena, transitioning from classical determinism to quantum probability, and the limitations of language in describing quantum phenomena, issues closely tied to the diverse interpretations of mathematical formalism [2]. Let us examine some of these aspects in more detail.

A lack of intuition regarding quantum phenomena compounds these difficulties. Our brains are evolutionarily adapted to make rapid decisions with minimal energy expenditure, often relying on the limbic system—an ability critical in dangerous situations. When confronted with events that appear illogical or contradictory, the anterior cingulate cortex

is activated, triggering other brain structures to seek explanations. This process leads to the formation of preconceived notions about the nature of physical phenomena [3]. While classical physics already generates a significant number of misconceptions [4], the list grows even longer for quantum physics [5]. In short, preconceptions are unavoidable, and any teaching or learning methodology must account for them, particularly in physics education.

The formal understanding of physical phenomena requires the establishment of neuronal pathways and the activation of the frontal lobe, particularly the dorsolateral prefrontal cortex [6], which entails substantial energy consumption. Moreover, the frontal lobe is the last brain region to undergo myelination, meaning that a certain level of brain maturity is required to comprehend abstract phenomena [7]. This indicates that the structure and functioning of the brain inherently influence how students learn physics, regardless of their geographic or cultural background. The formal learning of physics, therefore, necessarily demands effort [8].

Phenotypic differences between men and women suggest that while they may perform the same tasks, they often do so differently, utilizing distinct neural structures. For instance, the high connectivity in the female corpus callosum contributes to a more holistic cognitive approach [9]. From a purely cytoarchitectonic and physiological perspective, there is no reason why men and women should differ in their success levels for such tasks. Consequently, research on gender disparities has primarily focused on stereotypes rather than inherent abilities.

In higher education, difficulties in transferring mathematical knowledge often hinder the learning of physics. Motivation to pursue advanced mathematics at the university level is largely driven by satisfaction and achievement [10]. Many students, however, feel inadequately prepared for the level of mathematics required to study physics.

Reviewing previously learned material enhances knowledge retention and understanding, as demonstrated by Ebbinghaus forgetting curves [11]. Introducing aspects of quantum mechanics in pre-university courses could facilitate the gradual acquisition of knowledge.

A mechanistic approach to problem-solving often creates a false sense of understanding, necessitating alternative study methods for meaningful and holistic learning. Classical physics can serve as a bridge to quantum phenomena [12]. However, some authors [13] argue that prior knowledge can hinder the assimilation of new concepts in quantum mechanics due to the paradigm shift it entails. Thus, the effectiveness of a didactic strategy based on the classical–quantum continuity remains unclear in the context of quantum mechanics education.

Simulations have proven to be effective tools in facilitating physics learning [14], as has the incorporation of the history of science to address the conceptual challenges [15].

In any learning process, memory must be considered. The information we will retain in our memory starts in the entorhinal cortex (in the hippocampal gyrus) and goes to the hippocampus for a certain period of time, controlled by the prefrontal cortex, which coordinates working memory. This information is then sent to the temporal cortex (for long-term storage) and, through the fornix (septum), to the hypothalamus. Then, fibers exit to the mammillary bodies. Passing through the thalamus, the information is sent to the various cortical regions so that memories can be consciously evoked (explicit or declarative memory). The electrical signal goes also to the amygdala, where the affective imprint associated with the stimulus is generated, involving the hormone cortisol [16].

In any learning process, there is an affective factor. The reward and pleasure tract carries dopamine from the ventral tegmental area to the nucleus accumbens. We want to repeat an action when this tract is activated. It implies that motivations are needed when learning.

From a psychological standpoint, new information is integrated with existing knowledge frameworks [17]. Effective comprehension strategies include the use of practical applications, thought-provoking challenges, and maieutic methods to encourage discussions and confront difficulties as they arise [18]. These approaches leverage multiple motivational

factors: expectations [19], goal-setting [20], curiosity [21], and self-improvement [22]. These factors support the latest perspectives on motivation [23].

In Spain, there is a special physics course before starting university. Only a small part is related to 20th century physics. And not all students take it. In this course, some aspects of quantum mechanics are superficially treated, such as the quantization of energy, the photoelectric effect, or the de Broglie wavelength. There is a high mechanization of the problems in this part of the syllabus and little understanding of these phenomena.

We have observed that students have special difficulty in learning the following concepts when they begin to study quantum mechanics:

- The wave–particle duality of nature. A wavelength (wave behavior) can be associated with a particle with mass through the de Broglie wavelength:

$$\lambda = h/p \tag{1}$$

Their intuition about it is usually weak and the connection between the de Broglie wavelength and Heisenberg's uncertainty principle further complicates understanding it.

The mass–energy equivalence.

$$E = mc^2 \tag{2}$$

Although this expression belongs to relativity, its use in quantum physics is absolutely necessary. Students understand conservation of mass relatively well (e.g., the total mass of a cake's ingredients equals the mass of the baked cake). However, the understanding of mass–energy conservation is challenging. Although students apply the principle of energy conservation efficiently, they do not always really understand what they are doing, as the concept of energy is more abstract than the concept of mass and weight. Classical concepts are not always deeply internalized, which is why this proposal includes reviews of foundational ideas. Students find the mass–energy equivalence principle difficult to comprehend, often leading them to mechanize the use of mass units in electron volts (eV) without fully understanding their significance.

Statistical processes in quantum mechanics. Concepts such as stability, half-life, disintegration, and decay are far from classical determinism and intuition. The activity is intended to be an introduction to statistical processes. Thus, they become something more familiar to students.

Given these considerations, this research focuses on analyzing student motivations to engage with quantum physics. It emphasizes the patience, time, perseverance, and resilience required to master the subject, using real data from the Compact Muon Solenoid (CMS) at the European Organization for Nuclear Research (CERN) as a learning pathway. Starting from reality allows a better connection between the student and the subject. That is why it is so important to take advantage of the opportunity to learn with real data from CERN.

Although great progress has been made in understanding the brain processes of physics learning, there is a lack of didactic proposals based on the functioning of the brain. This article aims to contribute to the foundations of physics teaching.

## 2. Materials and Methods

Two tools were used for this study. The first one was a Likert-type questionnaire with a scale from 1 to 5 designed to measure students' motivations for learning quantum physics. The sample consisted of 81 students between 18 and 20 years old (N = 81, 36 women and 45 men), enrolled in the following degrees: artificial intelligence systems engineering, computer engineering, mathematical engineering, and architecture. Informed consent was given by all participants, participation was voluntary, and the questionnaire was completed anonymously.

The second tool was an activity designed for students.

A description of each tool is provided in the following sections.

### 2.1. Motivation Likert Questionnaire

The questionnaire was designed to measure the intrinsic motivation of the students. It consists of nine items:

Q1: I know applications of quantum physics to technology.

Q2: Quantum physics is a difficult subject to study.

Q3: I am interested in quantum physics.

Q4: It is interesting to learn using actual data from the CERN particle accelerator.

Q5: I find it useful to learn physics by identifying elementary particles in a real collider.

Q6: For me, it is a personal challenge to be able to analyze real data from CERN.

Q7: I am curious how data are obtained and analyzed in a real particle accelerator.

Q8: I find it motivating to learn from real data from the CERN accelerator.

Q9: My knowledge of classical physics would help me to understand quantum phenomena.

In the design of the questionnaire, the Delphi method was used, considering 3 professors specializing in quantum mechanics. These experts are university professors who are not related to this research work. After approval in two rounds of the questionnaire, two more experts (a doctor and a psychologist) considered it appropriate. The internal consistency of the test is validated through Cronbach's alpha ($\alpha = 0.848$).

Different statistical tests have been used in this study: normality tests (Kolmogorov–Smirnov and Shapiro–Wilk), a homoscedasticity test (Levene), and a non-parametric comparison test (Mann–Whitney U).

All analyses in this study were performed using IBM SPSS, Version 29.0.0.0 (241); SPSS Inc., Chicago, IL, USA.

### 2.2. CMS Activities

The whole activity is presented in Appendix A. In this section we want to develop the didactic reasons for its design. The activity begins with a description of the CMS structure and detectors. We want to engage our students by using current devices in cutting-edge research.

Some problems of classical physics are offered to establish a baseline of knowledge, revisiting some concepts. These problems will activate the dorsolateral region of the left frontal lobe, linked to logical–mathematical reasoning. All problems must be written and argued, offering a discussion of the results. The act of writing requires a prior organization of thought. This exercise improves connectivity between the hippocampus and neocortex.

All activities can be undertaken in small groups, so students will support each other and discuss what they are doing. Differences in neural plasticity and neural flexibility between students will mean that some of them will be able to solve problems more quickly. It is important that students support each other in their learning process. It takes time to delve into concepts, understand them, and create and reorganize neural tracts, avoiding misconceptions.

The most important and challenging activity is to determine the type of particle from the simulator traces (taken from real data). The activation of the amygdala allows us to associate the activity with the adventure of the pioneers in the study of elementary particles. It allows you to open your mind. It is exciting and motivating. Finding a particle among the proposed combinations is a challenge that invites students to repeat the action when they have achieved it for the first time. It is the sweet taste of success. It allows expectations to be met, produces personal satisfaction, and allows curiosity to be satisfied. It triggers dopaminergic activity, motivating to continue learning.

Using the fundamentals of quantum mechanics discussed above, such as the de Broglie wavelength, the statistical nature of quantum behavior, or the mass–energy relationship, is

now easier, since it is linked to an experience. Being part of a personal experience, it also improves its storage in declarative and long-term memories.

Even though this study is related to intrinsic motivations, the brain-based teaching approach has offered positive results in terms of learning physics [24].

## 3. Results

The direct results obtained in the motivation test are shown in Table 1. In the main sample there are 36 women and 45 men.

**Table 1.** Mean values and standard deviations obtained in the motivation test.

| Item | Mean | Std. Deviation |
|------|------|----------------|
| Q1 | 2.38 | 1.220 |
| Q2 | 4.16 | 0.968 |
| Q3 | 3.43 | 1.128 |
| Q4 | 3.78 | 1.183 |
| Q5 | 3.78 | 1.140 |
| Q6 | 3.90 | 1.136 |
| Q7 | 3.69 | 1.281 |
| Q8 | 3.73 | 1.118 |
| Q9 | 3.54 | 1.184 |

We are particularly interested in exploring potential motivational differences between men and women. Table 2 presents a comparison of response values based on gender.

**Table 2.** Mean values and standard deviations obtained in the motivation test by gender.

| Item | Gender | Mean | Std. Deviation | Std. Error Mean |
|------|--------|------|----------------|-----------------|
| Q1 | Female | 2.00 | 1.042 | 0.174 |
|    | Male   | 2.69 | 1.276 | 0.190 |
| Q2 | Female | 4.14 | 0.931 | 0.155 |
|    | Male   | 4.18 | 1.007 | 0.150 |
| Q3 | Female | 3.08 | 0.967 | 0.161 |
|    | Male   | 3.71 | 1.180 | 0.176 |
| Q4 | Female | 3.56 | 1.229 | 0.205 |
|    | Male   | 3.96 | 1.127 | 0.168 |
| Q5 | Female | 3.67 | 1.219 | 0.203 |
|    | Male   | 3.87 | 1.079 | 0.161 |
| Q6 | Female | 4.03 | 0.971 | 0.162 |
|    | Male   | 3.80 | 1.254 | 0.187 |
| Q7 | Female | 3.64 | 1.246 | 0.208 |
|    | Male   | 3.73 | 1.321 | 0.197 |
| Q8 | Female | 3.64 | 0.990 | 0.165 |
|    | Male   | 3.73 | 1.217 | 0.181 |
| Q9 | Female | 3.67 | 0.986 | 0.164 |
|    | Male   | 3.44 | 1.324 | 0.197 |

In order to compare both samples, the normality study was carried out, as shown in Table 3. Both parametric and non-parametric tests indicate that the samples do not have Gaussian distributions.

**Table 3.** Normality test in Male–Female samples.

| Item | Gender | Kolmogorov–Smirnov * | | | Shapiro–Wilk | | |
|---|---|---|---|---|---|---|---|
| | | Statistic | df | Sig. | Statistic | df | Sig. |
| Q1 | Female | 0.222 | 36 | 0.000 | 0.835 | 36 | 0.000 |
| | Male | 0.159 | 45 | 0.006 | 0.894 | 45 | 0.001 |
| Q2 | Female | 0.267 | 36 | 0.000 | 0.812 | 36 | 0.000 |
| | Male | 0.260 | 45 | 0.000 | 0.760 | 45 | 0.000 |
| Q3 | Female | 0.216 | 36 | 0.000 | 0.908 | 36 | 0.006 |
| | Male | 0.196 | 45 | 0.000 | 0.855 | 45 | 0.000 |
| Q4 | Female | 0.186 | 36 | 0.003 | 0.883 | 36 | 0.001 |
| | Male | 0.249 | 45 | 0.000 | 0.806 | 45 | 0.000 |
| Q5 | Female | 0.191 | 36 | 0.002 | 0.865 | 36 | 0.000 |
| | Male | 0.209 | 45 | 0.000 | 0.860 | 45 | 0.000 |
| Q6 | Female | 0.239 | 36 | 0.000 | 0.830 | 36 | 0.000 |
| | Male | 0.230 | 45 | 0.000 | 0.810 | 45 | 0.000 |
| Q7 | Female | 0.253 | 36 | 0.000 | 0.862 | 36 | 0.000 |
| | Male | 0.209 | 45 | 0.000 | 0.829 | 45 | 0.000 |
| Q8 | Female | 0.226 | 36 | 0.000 | 0.891 | 36 | 0.002 |
| | Male | 0.232 | 45 | 0.000 | 0.844 | 45 | 0.000 |
| Q9 | Female | 0.271 | 36 | 0.000 | 0.859 | 36 | 0.000 |
| | Male | 0.169 | 45 | 0.002 | 0.881 | 45 | 0.000 |

* Lilliefors significance correction.

The Levene test for homoscedasticity indicates that the sample variances are unequal, except for items Q1, Q3, and Q9.

Given the characteristics of the samples and the robustness of non-parametric tests, the Mann–Whitney U test is the most appropriate method for comparing responses by gender. The obtained results are presented in Table 4.

**Table 4.** Independent-samples (Male–Female) Mann–Whitney U test.

| Null Hypothesis | Sig. [a,b] |
|---|---|
| The distribution of Q1 is the same across categories of Gender. | 0.013 |
| The distribution of Q2 is the same across categories of Gender. | 0.728 |
| The distribution of Q3 is the same across categories of Gender. | 0.008 |
| The distribution of Q4 is the same across categories of Gender. | 0.113 |
| The distribution of Q5 is the same across categories of Gender. | 0.517 |
| The distribution of Q6 is the same across categories of Gender. | 0.572 |
| The distribution of Q7 is the same across categories of Gender. | 0.629 |
| The distribution of Q8 is the same across categories of Gender. | 0.296 |
| The distribution of Q9 is the same across categories of Gender. | 0.556 |

[a] The significance level is 0.050. [b] Asymptotic significance is displayed.

There are only statistically significant differences in items Q1 and Q3.

## 4. Discussion

The discussion of results is offered in the order of items in the questionnaire.

Item Q1 reveals that not all students have a clear understanding of the technological applications of quantum physics, particularly women. The differences between men and women are statistically significant. A plausible explanation could be that men generally exhibit greater interest in technological devices compared to women [25,26]. Consequently, a key takeaway is that teachers should emphasize technological applications to engage students and spark interest in the subject.

There is broad consensus among students that quantum physics is a challenging subject to study [27,28], as reflected in the mean response for Q2. This item serves as a control metric for responses across the entire questionnaire, as the intrinsic difficulty of studying quantum mechanics has been discussed in the introduction. The brain is naturally adapted for intuitive knowledge of the physical world, whereas formal learning requires strategies to establish new neural pathways that facilitate learning. These strategies include patience, perseverance, critical reflection, and practice. As such, Q2 provides a baseline for determining reasonable standard deviations across the other items.

Item Q3 offers an interesting result: women show statistically significantly less interest in quantum mechanics compared to men. A probable reason for this could be the historically lower representation and perceived role of women in physics [29].

Items Q4–Q8 provide insights into the motivational interests we aim to measure in this study, aligning with the classic theories of motivation outlined in the introduction.

Students positively value the use of real data from CERN, not only for learning physics but also for gaining experience in statistical processing of large datasets. This is particularly evident among men, though no statistically significant gender difference is observed, as reflected in Q4 and Q5.

The proposed activity is considered a motivating challenge (Q6), with women expressing slightly higher appreciation. However, there are no statistically significant differences between gender groups.

An overall analysis of the results from items Q1–Q3 has suggested that women are less familiar with technological applications related to quantum mechanics compared to men and are more likely to perceive the subject as highly challenging. Consequently, it is reasonable to infer that women may feel less motivated to study quantum physics than men. However, this result contrasts with the higher motivation reported by women in item Q6, which relates the challenge as a motivational factor. This may reflect the growing confidence women are gaining in science when they really face the subject or their strong sense of responsibility and commitment.

It is common for women to feel less capable in physics or mathematics, especially when they are in the minority within a class. Avoiding gender-imbalanced groups [30] is a plausible explanation for the motivational patterns observed in this activity. In our sample, 44% of participants are women, and 56% are men.

Students express curiosity about how data are analyzed to draw conclusions in quantum mechanics (Q7), with only minor differences between men and women.

Both groups find learning from CMS data to be motivating, with no significant differences between them (Q8).

Regarding prior knowledge, both groups believe their understanding of classical physics will be helpful for learning quantum mechanics (Q9). However, this belief cannot be fully validated, as no learning outcomes were measured in this study. Although the differences are not statistically significant, women appear to have slightly more confidence in the usefulness of their classical physics knowledge compared to men.

We also decided to compare the results from different academic degrees, but it does not reveal significant differences, as indicated by the Kruskal–Wallis test with Bonferroni corrections. The only exceptions are observed for students in mathematical engineering,

who show statistically significant differences compared to architecture students for Q1 ($p = 0.07$) and compared to computer engineering students for Q5 ($p = 0.011$). Overall, it can be concluded that students across these technical degrees share similar motivations for studying quantum mechanics.

The overall results are encouraging, as indicated by the mean responses. Students exhibit strong intrinsic motivation towards the activity. The proposed activity aligns well with the motivational theories discussed earlier, fostering engagement through challenge, curiosity, expectations, and personal growth.

The lack of significant differences between gender groups is particularly promising. While women express slightly less interest in quantum mechanics, the activity inspires comparable levels of motivation in both men and women. This makes it a valuable tool for addressing gender stereotypes [31]. Although much work remains to ensure equitable inclusion of women in physics and mathematics, progress has been made in recent years. This study aims to contribute to these advancements, encouraging women to take on increasingly significant roles in these fields. As described above, men and women perform the same tasks, with the same success, but not always using the same brain structures. Therefore, it is important to find activities that are equally motivating for men and women and to avoid neglecting anyone.

This study only analyzes student motivations. Future research should investigate the impact of this activity on students' knowledge of quantum mechanics, assessing whether it effectively enhances their understanding of the subject.

## 5. Conclusions

Pedagogical applications of brain knowledge are important for learning physics and can be implemented in colleges and universities. An activity utilizing real data from the CMS proves to be particularly motivating for university students. This approach introduces concepts of quantum mechanics through the lens of challenge, curiosity, expectations, and personal growth. The proposal has been equally successful for men and women, with no statistically significant differences between their intrinsic motivations, although women are not as interested in quantum mechanics and technological devices as men. The desire to learn fostered by this activity make it a valuable tool for encouraging women to engage in the study of quantum physics, helping to bridge the existing gender gap in this technical field. Future research should offer some results on the effectiveness of physics learning using the proposed activity.

**Author Contributions:** Conceptualization, H.R.-M. and M.A.-H.; methodology, H.R.-M.; software, H.R.-M.; validation, H.R.-M.; formal analysis, H.R.-M.; investigation, H.R.-M.; resources, H.R.-M.; data curation, H.R.-M.; writing—original draft preparation, H.R.-M.; writing—review and editing, H.R.-M. and M.A.-H.; visualization, H.R.-M. and M.A.-H.; supervision, H.R.-M. and M.A.-H.; project administration, H.R.-M. and M.A.-H. All authors have read and agreed to the published version of the manuscript.

**Funding:** This research received no external funding.

**Data Availability Statement:** The raw data supporting the conclusions of this article will be made available by the authors on request.

**Conflicts of Interest:** The authors declare no conflicts of interest.

## Appendix A

1. Objectives.

The main objectives of the activity are:

Identify the Lorentz force applied to charged particles.

Application of Ampère's law to a solenoid.

Determine the type of particle obtained in a collision in the CMS detector and understand the importance of stability and half-life of a particle.

Relate the experience to the standard model.

Calculate and explain the meaning of the De Broglie wavelength.

Explain the mass–energy bond.

Link the experience with quantum mechanics and electromagnetic fields.

2.　CMS description.

Each part of the detector is used for the detection of different particles. The internal structure of the CMS is as shown in Figure A1.

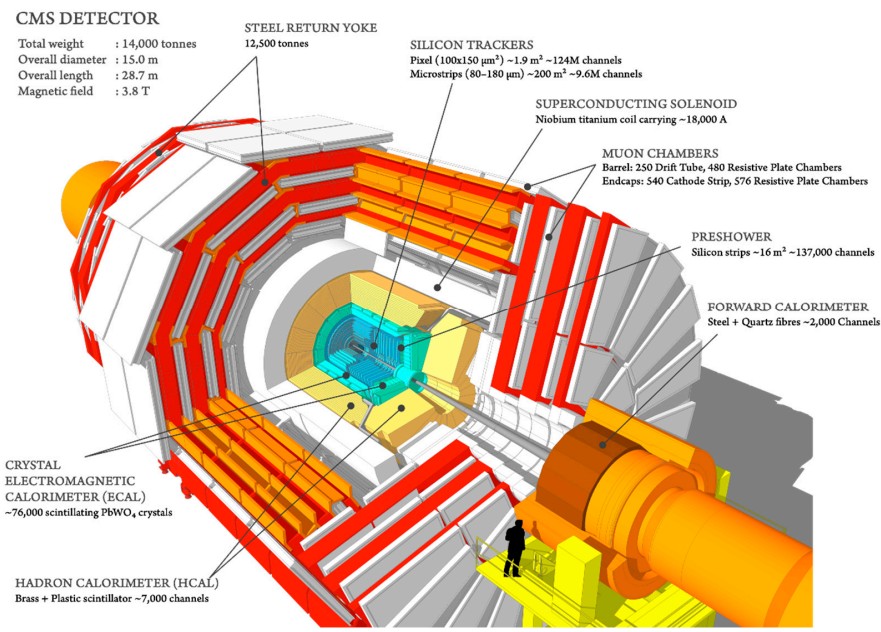

**Figure A1.** An overview of the CMS detector [32].

Different detectors can be visible in the simulator and various particles can be detected, depending on the CMS area, as shown in Figure A2.

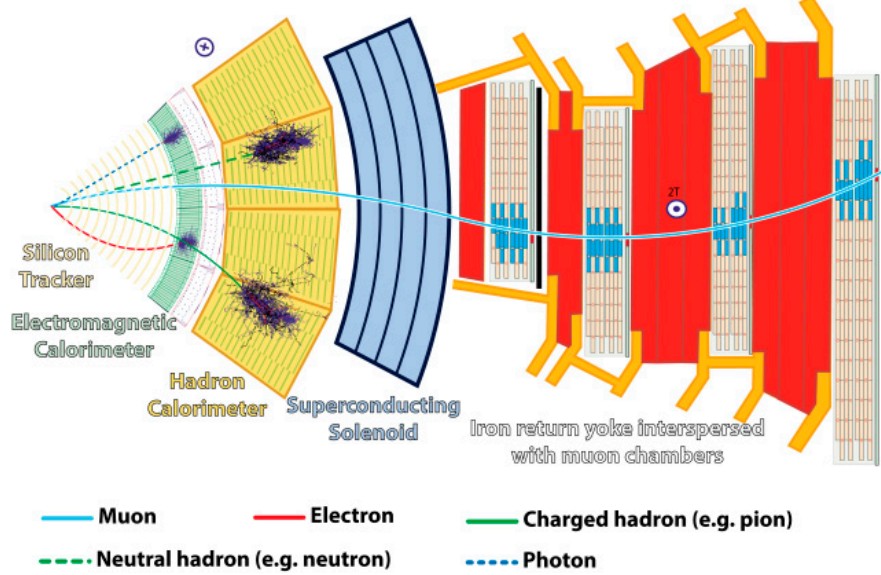

**Figure A2.** Cross-sectional view of the CMS detector [33].

Bosons have a short half-life time, so they will be detected indirectly through their decay to other particles, as shown in Figure A3.

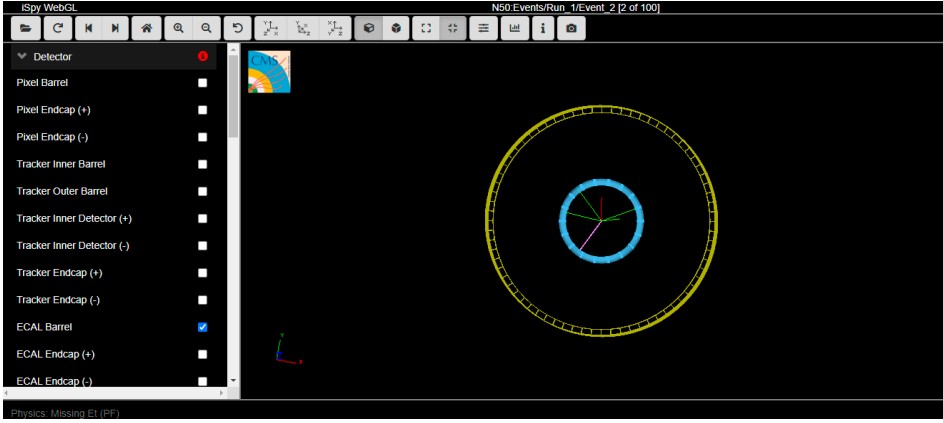

**Figure A3.** Initial and possible final states.

3.    Software and instructions.

To begin, open the simulator [34,35] and download the corresponding data sheets [36]. Once the simulator is open, load the data from the library labeled N50/(navigate to Open File > Open File from the Web > N50/). Click on any event to load all events from the library. If the first link does not work, use the second link and load the events from masterclass_1. Select 30 events for analysis.

As shown in Figure A4, several configurable tabs will appear on the left side of the screen, while the event to be analyzed will be displayed on the right.

**Figure A4.** Simulator screenshot.

Next, there is a brief description of the main detectors needed:

*Detector*: you must activate ECAL Barrel and HCAL Outer only.

*Physics*: you must activate Electron Tracks, Tracker Muons, Global Muons, and Missing Et. You can try activating other options, especially the *Physics tab*.

All other options should be turned off, avoiding possible confusion and having a clear view of the simulator.

To identify the particles, you need to know two things:

- The color of the trace provided by the simulator:

  Red: muon

Green: electron (or anti-electron)
Pink: stray energy. It will be associated with a neutrino.
Yellow: photon

- Direction of rotation or deviation of the particle:

Dexentropic (clockwise) direction: positively charged particle.
Leavotropic (counterclockwise) direction: negatively charged particle.
The direction of spin will determine whether it is a particle or an antiparticle. For example, the electron and positron will spin in opposite directions.
We will indicate the particles according to the associated letter: electron (e), muon ($\mu$), neutrino ($\nu$).

4. Activities to solve.
   The proposed activities to be solved by the students were the following:

   4.1 Based on the analysis of the particles and their curvature in the presence of a magnetic field, determine the direction and orientation of the magnetic field. Sketch the detector and the magnetic field.

   4.2 Provide a brief description of the Standard Model of particle physics, highlighting its most notable features, such as particle mass, charge, and spin.

   4.3 CERN's magnetic field strength currently reaches 8.33 T with current intensities of 11,850 A. Calculate the number of turns a solenoid with a diameter of 15 m would require to achieve this field strength.

   4.4 From the final states visible in the simulator, deduce the initial states of the events.

   4.5 Investigate whether identical final states could originate from different initial states and connect this observation to the concept of probability.

   4.6 The Higgs boson has an approximate mass of 125 GeV. Determine how many times more massive the Higgs boson is compared to an electron.

   4.7 Select an event involving a clearly curved muon. Estimate the radius of curvature (consider that the electron detector has a radius of approximately 1.5 m). Given that the muon's mass is approximately 106 MeV, its charge equals that of the electron, and the magnetic field strength is 3.8 T, calculate the muon's velocity.

   4.8 Using the velocity calculated in the previous step, calculate the De Broglie wavelength associated with the muon.

   4.9 Determine the electric field strength necessary to prevent the muon from being deflected.

   4.10 Relate this practical activity to the theoretical principles required to complete it successfully.

   Estimated time needed: 4 h.

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
