# Peer review of "An Introduction to Quantum Mechanics Through Neuroscience and CERN Data"

_quantumrep, doi:10.3390/quantum7010005_

Round 1
Reviewer 1 Report
Comments and Suggestions for Authors
This brief paper is an attempt to familiarise the basic tenets of quantum mechanics with the technicalities of the results obtained at CERN. To this end the materials and methods are formulated in the form of a questionnaire. Then the objectives of CMS activities are summarised in which an overview of the CMS detector is given with an explanation of Initial and possible final states. There is another table in which direct results obtained in the motivation test are given. Finally, some discussions and a conclusion is provided. The paper marginally qualifies to be published in Quantum Reports.
Author Response
Thanks for your kind comments.
Reviewer 2 Report
Comments and Suggestions for Authors
This article deals with an important aspect of teaching quantum mechanics: the lack of daily experience and intuition. This lack is one of the biggest difficulties that students of quantum mechanics encounter when they face the subject. This work presents a holistic approach for introducing some basic concepts of quantum mechanics. Their claim is that it is possible to learn this part of physics with real open data obtained at CERN, involving classic concepts of dynamics and electromag-netism and linking them to their quantum counterparts, understanding their similarities and differences. Furthermore, they relate such activity is to our knowledge of the brain and the way it learns physics, considering the so called cytoarchitectonic regions and the psychological processes in-volved. New advances in neuroscience allow us to design activities taking the student into account but without neglecting the rigor and precision of the pertinent abstract concepts.
Intrinsic motivations have been measured using a Likert-type test to 81 un-dergraduate students from artificial intelligence systems engineering, computer engineering, mathematical engineering and architecture. The results obtained show that the above described approach enhances the intrinsic motivation of students towards the study of quantum mechanics. The motivation is enhanced in both men and women.
This paper presents a novel approach to addressing a common challenge in teaching quantum mechanics: the lack of intuitive understanding among students. By proposing a holistic activity that links classical physics concepts to their quantum counterparts, the authors aim to enhance engagement and motivation, particularly among undergraduate students. The integration of real open data from CERN and considerations from neuroscience adds depth and relevance to the study.
The issue of students struggling with abstract concepts in quantum mechanics is well recognized. The paper's focus on enhancing intuition through tangible data is a commendable approach.
Also, by incorporating concepts from dynamics, electromagnetism, and neuroscience, the paper´s proposed activity promotes a comprehensive understanding of physics. This interdisciplinary method is likely to resonate with students from various engineering and architectural backgrounds.
Overall, this paper offers a promising and innovative approach to improving student motivation and understanding in quantum mechanics. This work could significantly contribute to educational practices in physics. I recommend acceptance.
cc
Author Response
Thanks for your kind comments.
Reviewer 3 Report
Comments and Suggestions for Authors
Dear Authors,
Thank you for submitting your manuscript, An Introduction to Quantum Mechanics through Neuroscience and CERN Data. I appreciate the innovative approach you have taken to connect quantum mechanics, neuroscience, and CERN data, and your work has the potential to contribute significantly to the field. However, there are several areas where revisions and clarifications could strengthen the manuscript.
Abstract
This abstract presents an interesting approach to teaching quantum mechanics, but it could be strengthened with some revisions. The abstract mentions an "activity" but doesn't explicitly state the purpose of the article. Is it to describe the activity, report on its effectiveness, or propose a new pedagogical approach? Start with a clear statement of purpose. What makes this approach unique? Emphasise the connection between quantum mechanics, CERN data, and neuroscience. Briefly elaborate on how neuroscience is incorporated. Which "cytoarchitectonic regions" and psychological processes are considered?
Methodology: Provide more context for the study. How was the activity implemented? What specific aspects of intrinsic motivation were measured?
Conclusion: The statement about enhancing the role of women in physics feels tacked on. Either remove it or provide more context. If the study specifically addressed gender differences in motivation, briefly mention the findings
Keywords: These are not aligned with the topic, for example, quantum physics and in the topic Quantum Mechanics, Neuroscience is also missing in the keywords
Introduction
Your introduction lays a strong foundation but could benefit from more focus and cohesion.
- Streamline your discussion to highlight the most relevant aspects of your research. For example, the discussion of gender differences (lines 53–58) feels tangential and could be condensed or removed.
- Strengthen the connection between CERN data and the challenges outlined earlier in the introduction.
- Clearly articulate the gap in existing literature and how your research addresses this gap.
- Break down longer sentences (e.g., lines 31–35) for better readability.
Materials and Methods
This section is well-structured but requires additional detail for clarity and reproducibility.
- Participant Details: Provide more context about the participants, such as demographics or prior exposure to quantum mechanics.
- Questionnaire Development: Expand on the Delphi method—how many rounds were conducted, and how were experts selected?
- CMS Activity:
- Specify the CERN data used and how it was accessed.
- Detail the format, duration, and resources for the activity.
- Include a step-by-step procedure for the activity.
- Data Analysis: Clarify how quantitative and qualitative data were analyzed. Specify statistical methods and qualitative analysis techniques.
- Ethical Considerations: Include details about consent and data privacy.
Results and Discussion
The results section is well-supported with tables, but the discussion requires better focus and depth.
- Streamline the discussion to present a clear narrative. Avoid jumping between unrelated points.
- Avoid overinterpretation of results; ensure conclusions are fully supported by the data.
- Use precise language instead of vague terms like "in a statistically significant way."
- Balance the focus on gender differences with other insights from the study.
Conclusion
Your conclusion could better highlight the study's unique contributions and practical implications.
- Expand on the broader implications of your findings beyond gender differences.
- Include specific recommendations for future research and practical applications.
We look forward to your revised submission and are confident these revisions will enhance the clarity and impact of your work. If you have any questions or need further clarification, please do not hesitate to contact us.
Best regards,
Comments on the Quality of English LanguageGood proofreading is required. Some sentences are long and some are very short.
Author Response
Thanks for your comments. We are sure that the article is better thanks to your suggestions.
Abstract
This abstract presents an interesting approach to teaching quantum mechanics, but it could be strengthened with some revisions. The abstract mentions an "activity" but doesn't explicitly state the purpose of the article. Is it to describe the activity, report on its effectiveness, or propose a new pedagogical approach? Start with a clear statement of purpose. What makes this approach unique? Emphasise the connection between quantum mechanics, CERN data, and neuroscience. Briefly elaborate on how neuroscience is incorporated. Which "cytoarchitectonic regions" and psychological processes are considered?
Methodology: Provide more context for the study. How was the activity implemented? What specific aspects of intrinsic motivation were measured?
Conclusion: The statement about enhancing the role of women in physics feels tacked on. Either remove it or provide more context. If the study specifically addressed gender differences in motivation, briefly mention the findings
Abstract re-elaborated in the manuscript.
Keywords: These are not aligned with the topic, for example, quantum physics and in the topic Quantum Mechanics, Neuroscience is also missing in the keywords
Changed to: quantum physics learning; motivation; physical particles; women in physics; CERN; CMS, neuroscience.
Introduction
Your introduction lays a strong foundation but could benefit from more focus and cohesion.
- Streamline your discussion to highlight the most relevant aspects of your research. For example, the discussion of gender differences (lines 53–58) feels tangential and could be condensed or removed.
- Strengthen the connection between CERN data and the challenges outlined earlier in the introduction.
- Clearly articulate the gap in existing literature and how your research addresses this gap.
- Break down longer sentences (e.g., lines 31–35) for better readability.
Very useful tips. We have developed the introduction.
Materials and Methods
This section is well-structured but requires additional detail for clarity and reproducibility.
- Participant Details: Provide more context about the participants, such as demographics or prior exposure to quantum mechanics.
It has been included in the article:
- In Spain, there is a special physics course before starting university. Only a small part is related to 20th century physics. And not all students take it. In this course, some aspects of quantum mechanics are superficially treated, such as the quantization of energy, the photoelectric effect or the de Broglie wavelength. There is a high mechanization of the problems in this part of the syllabus and little understanding of these phenomena.
It has been also included a brief comment of each difficulty.
- There are 36 women and 45 men in the sample.
- The sample consisted of 81 students between 18 and 20 years old.
- Questionnaire Development: Expand on the Delphi method—how many rounds were conducted, and how were experts selected?
It has been included in the article, in the description of the Likert:
In the design of the questionnaire, the Delphi method was used, considering 3 professors specializing in quantum mechanics. These experts are university professors who are not related to this research work. After approval in two rounds of the questionnaire, two more experts (a doctor and a psychologist) considered it appropriate. The internal consistency of the test is validated through Cronbach's alpha (α = 0.848).
- CMS Activity:
- Specify the CERN data used and how it was accessed.
- Detail the format, duration, and resources for the activity.
- Include a step-by-step procedure for the activity.
It is all described in the “CMS activity” section, but duration: it takes 4 hours to complete the activity. Added in the article, at the end of the activity description.
- Data Analysis: Clarify how quantitative and qualitative data were analyzed. Specify statistical methods and qualitative analysis techniques.
It is deeply explained in “results”, both statistical methods and qualitative analysis techniques. Added in the description of the sample (“methods”).
- Ethical Considerations: Include details about consent and data privacy.
It has been included in the article (in the description of the sample):
Informed consent was given by all participants, participation was voluntary and the questionnaire was completed anonymously.
Results and Discussion
The results section is well-supported with tables, but the discussion requires better focus and depth.
- Streamline the discussion to present a clear narrative. Avoid jumping between unrelated points.
We have restructured the discussion in order to be clearer and consistent.
- Avoid overinterpretation of results; ensure conclusions are fully supported by the data.
- Use precise language instead of vague terms like "in a statistically significant way."
Revision of the English done.
- Balance the focus on gender differences with other insights from the study.
Added, lines 289-292.
Conclusion
Your conclusion could better highlight the study's unique contributions and practical implications.
- Expand on the broader implications of your findings beyond gender differences.
- Include specific recommendations for future research and practical applications.
We have improved this section from your comments.
Reviewer 4 Report
Comments and Suggestions for Authors
This is an interesting paper which provides some useful insights for teachers in physics and especially for those trying to increase the involvement of women in physics.
Unfortunately as it stands the paper needs a major revision to make it more rigorous and logical. The introduction is very weak. There is an extensive literature on the learning of scientific concepts and in particular physics concepts. Some of this goes back to Piaget but there is lots of more modern literature. The references to neurophysiology are very superficial to say the least - they place an emphasis on neuroanatomy - even that is superficial, but they ignore the extensive literature on embodied cognition which plays a role in creating the insight associated with classical physics. Suitable exposure to quantum processes from an early age could provide the kinds of insight that they refer to here.
Some statements are confusing. For example "Preconceived ideas are then formed about what the explanations of different physical phenomena should be like". Preconceived ideas might be elicited but surely not formed at the time of explanation - in that case they would not be preconceived.
This statement "Our brain is prepared to make quick decisions with little expenditure of energy, through the limbic system. Our lives can depend on it in a dangerous situation. When an event seems illogical or contradictory to us, the anterior cingulate is activated, which demands the intervention of brain structures that offer us explanations" is full of partial truths but seems irrelevant for learning physics.
In this sentence "The formal understanding of physical phenomena requires the establishment of neuronal tracts and the activation of the frontal lobe, especially the dorsolateral region, with what this means in terms of energy consumption. " nothing really new is said. Any new learning of any kind requires changes in the brain. There is no special relationship between this and learning physics, or energy consumption.
"Furthermore, the frontal lobe is the last structure to become myelinated, so a certain brain maturity is required to understand such abstract phenomena [7].
This is not a direct logical conclusion.
"In other words, the way the brain works conditions the learning of physics, regardless of the part of the world in which the students are located. "
The way the brain works conditions everything that the brain learns. It is not clear exactly how this impacts the learning of physics.
"Formal learning of physics necessarily requires effort" I think that pretty much all formal learning requires effort.
The authors suggest something new in education can be drawn from neurobiology, but this not not appear anywhere in what follows in the paper. The questionnaire does not use any neurobiological knowledge per se. It is a pretty standard self report questionnaire. There is little discussion of the analysis of the validity and reliability of the questionnaire, which is basic in psychology. The alpha Cronbach alone is not sufficient to convince one of this. If the test is not sufficiently valid and reliable, then the results cannot be considered as such either.
The entire section describing from the start of 2.2 to section 3 can be removed. It is never referred to in the actual study, nor in the results or discussion. It is referred to in the questionnaire but adds little to the paper. Reports related to the students actually working with the simulation are not provided - I see no need for this material - if they really wish to describe it, then it could be placed in an appendix.
The English language certainly requires some improvement.
I think that the results are interesting, but more work is needed - a major revision is required.
Comments on the Quality of English Language
The paper does need a thorough read through for English usage. There are lots of awkward or agrammatical phrasings throughout.
Author Response
Thanks for your comments. We are sure that the article is better thanks to your suggestions.
This is an interesting paper which provides some useful insights for teachers in physics and especially for those trying to increase the involvement of women in physics.
Unfortunately as it stands the paper needs a major revision to make it more rigorous and logical. The introduction is very weak. There is an extensive literature on the learning of scientific concepts and in particular physics concepts. Some of this goes back to Piaget but there is lots of more modern literature. The references to neurophysiology are very superficial to say the least - they place an emphasis on neuroanatomy - even that is superficial, but they ignore the extensive literature on embodied cognition which plays a role in creating the insight associated with classical physics. Suitable exposure to quantum processes from an early age could provide the kinds of insight that they refer to here.
We cannot see this point very clearly. What is said about Piaget? What is contradictory regarding more recent studies? Would you be so kind as to be more specific, please?
Each anatomical structure mentioned is also described in physiological terms. The relationship is also offered through references.
We have added some information about the neurological tracts in a general learning process (Swaab reference).
Some statements are confusing. For example "Preconceived ideas are then formed about what the explanations of different physical phenomena should be like". Preconceived ideas might be elicited but surely not formed at the time of explanation - in that case they would not be preconceived.
Preconceived ideas are prior to the formal explanation of a natural phenomenon. Written again, in order to be clearer.
This statement "Our brain is prepared to make quick decisions with little expenditure of energy, through the limbic system. Our lives can depend on it in a dangerous situation. When an event seems illogical or contradictory to us, the anterior cingulate is activated, which demands the intervention of brain structures that offer us explanations" is full of partial truths but seems irrelevant for learning physics.
It is not irrelevant, as it is important for any learning process, the way the brain thinks. It is very well described in Thinking, Fast and Slow, written by Daniel Kahneman.
In this sentence "The formal understanding of physical phenomena requires the establishment of neuronal tracts and the activation of the frontal lobe, especially the dorsolateral region, with what this means in terms of energy consumption. " nothing really new is said. Any new learning of any kind requires changes in the brain. There is no special relationship between this and learning physics, or energy consumption.
Formal learning of physics requires the frontal lobe. Other kind of learning does not imply preconception and/or misconceptions. Other learning does not directly or so substantially require the dorsolateral region of the brain. A mechanical activity will demand the activation of the basal ganglia or the cerebellum, for example. A different situation: learning formal music (from the classic period, XVII-XIX centuries) does not requires to get over preconceptions or misconceptions.
"Furthermore, the frontal lobe is the last structure to become myelinated, so a certain brain maturity is required to understand such abstract phenomena [7].
This is not a direct logical conclusion.
It is not a conclusion, but a proven fact. Let's try to teach classical physics to a 4-year-old child. Most likely, we will not succeed… Revised wording in order to be clearer.
"In other words, the way the brain works conditions the learning of physics, regardless of the part of the world in which the students are located. "
The way the brain works conditions everything that the brain learns. It is not clear exactly how this impacts the learning of physics.
Learning physics requires changing patterns and preconceptions. This does not happen in all learning processes. Revised wording in order to be clearer.
"Formal learning of physics necessarily requires effort" I think that pretty much all formal learning requires effort.
May be, but in learning physics has been proven. Not all formal learning requires breaking down the prejudices generated by the brain. See Posner reference.
The authors suggest something new in education can be drawn from neurobiology, but this not not appear anywhere in what follows in the paper. The questionnaire does not use any neurobiological knowledge per se. It is a pretty standard self report questionnaire. There is little discussion of the analysis of the validity and reliability of the questionnaire, which is basic in psychology. The alpha Cronbach alone is not sufficient to convince one of this. If the test is not sufficiently valid and reliable, then the results cannot be considered as such either.
Neuroimaging resources are usually expensive. Once the areas associated with an activity have been established, it would not be necessary to subject students to this type of test. This is done with the Stroop test, Wisconsin cards or any ADHD dissorder detection test. It is not necessary to check the dopamine level between the ventral tegmental area and the nucleus accumbens. Nor to determine dementia due to Lewy bodies or Alzheimer's. The anamnesis is enough in psychology and psychiatry. We use these kinds of tests.
Statistical reliability and approval by experts is usually sufficient in a test like the one shown in the article.
It has been reinforced the description of the sample and the test.
To make the relationship between activity and neuroscience clearer, we have linked activity to the points made in the introduction.
The entire section describing from the start of 2.2 to section 3 can be removed. It is never referred to in the actual study, nor in the results or discussion. It is referred to in the questionnaire but adds little to the paper. Reports related to the students actually working with the simulation are not provided - I see no need for this material - if they really wish to describe it, then it could be placed in an appendix.
It is important for us that the activity appears in the article, as it may be useful to other professors.
The English language certainly requires some improvement.
We have done a revision of the English.
Round 2
Reviewer 3 Report
Comments and Suggestions for Authors
Dear Authors,
Thanks for addressing most of the comments.
Your manuscript is ready to go and I do not require any further improvements.
Kind regards
Comments on the Quality of English LanguageSatisfactory but recommend a good proofreading before submission